In silico analysis of maize HDACs with an emphasis on their response to biotic and abiotic stresses

Zhang Kang 1 2
Yu Lu 1 2
Pang Xi 1 2
Cao Hongzhe 1 2
Si Helong 1 2
Zang Jinping 1 2
Xing Jihong xingjihong@hebau.edu.cn xingjihong2000@126.com 1 2
Dong Jingao dongjingao@126.com 1 2
1 College of Life Science, Hebei Agricultrual University , Baoding , Hebei , China
2 Hebei Key Laboratory of Plant Physiology and Molecular Pathology, Hebei Agricultrual University , Baoding , Hebei , China
Sobral Mar
Electronic publication date: 2020 Feb 12
Publication date: 2020
Volume: 8
Electronic Location ID: e8539
Received 2019 Jul 4; Accepted 2020 Jan 9
Copyright: ©2020 Zhang et al.
Copyright year: 2020
Copyright holder: Zhang et al.
License: This is an open access article distributed under the terms of the Creative Commons Attribution License, which permits unrestricted use, distribution, reproduction and adaptation in any medium and for any purpose provided that it is properly attributed. For attribution, the original author(s), title, publication source (PeerJ) and either DOI or URL of the article must be cited.
License URL: https://creativecommons.org/licenses/by/4.0/

Keywords: Histone deacetylase, Maize, Phylogenetic relationship, Expression, Stress response

Funding: National Key Research and Development Program of China 2016YFD0300704 National Natural Science Foundation of China 31901864 Natural Science Foundation of Hebei Province C2019204141 China Agriculture Research System CARS-02-25 Starting Grant from Hebei Agricultural University ZD201721 This work was supported by the National Key Research and Development Program of China (2016YFD0300704), the National Natural Science Foundation of China (31901864), the Natural Science Foundation of Hebei Province (C2019204141), the China Agriculture Research System (CARS-02-25), and a Starting Grant from Hebei Agricultural University (ZD201721). The funders had no role in study design, data collection and analysis, decision to publish, or preparation of the manuscript.

==============================
Histone deacetylases (HDACs) are key epigenetic factors in regulating chromatin structure and gene expression in multiple aspects of plant growth, development, and response to abiotic or biotic stresses. Many studies on systematic analysis and molecular function of HDACs in Arabidopsis and rice have been conducted. However, systematic analysis of HDAC gene family and gene expression in response to abiotic and biotic stresses has not yet been reported. In this study, a systematic analysis of the HDAC gene family in maize was performed and 18 ZmHDACs distributed on nine chromosomes were identified. Phylogenetic analysis of ZmHDACs showed that this gene family could be divided into RPD3/HDA1, SIR2, and HD2 groups. Tissue-specific expression results revealed that ZmHDACs exhibited diverse expression patterns in different tissues, indicating that these genes might have diversified functions in growth and development. Expression pattern of ZmHDACs in hormone treatment and inoculation experiment suggested that several ZmHDACs might be involved in jasmonic acid or salicylic acid signaling pathway and defense response. Interestingly, HDAC genes were downregulated under heat stress, and immunoblotting results demonstrated that histones H3K9ac and H4K5ac levels were increased under heat stress. These results provide insights into ZmHDACs, which could help to reveal their functions in controlling maize development and responses to abiotic or biotic stresses.

Introduction

Dynamic chromatin structures have primary importance in modulating gene activities in higher eukaryotes (Luger, Dechassa & Tremethick, 2012). The chromatin structure can be affected by histone modifications, DNA methylation, and chromatin remodeling (Allis & Jenuwein, 2016). Among histone modifications, histone acetylation is one of the most widely studied (Shahbazian & Grunstein, 2007). Histone acetylation is generally associated with a chromatin structure that is open and therefore accessible to transcription factors or transcription regulators (Liu et al., 2014). The level of histone acetylation is regulated by histone acetyltransferases (HATs) and histone deacetylases (HDACs) (Peserico & Simone, 2011).

HDACs are a class of enzymes that remove acetyl groups from core histones (H2A, H2B, H3, and H4), thereby regulating gene expression. They are widely distributed in animals, yeasts, and plants. Based on homology analysis of yeast HDACs sequences, HDACs in plants can be grouped into three different families: the RPD3/HDA1, SIR2, and HD2 family (Pandey et al., 2002). RPD3/HDA1 is the largest family of HDACs, with members that are homologous to yeast RPD3 and HDA1 and contain a typical HDAC domain (Yang & Seto, 2008). The members of the SIR2 family are conserved from prokaryotes to eukaryotes, and use NAD+ as a coenzyme to exercise HDACs activity (Imai et al., 2000). HD2 family members were first identified in maize and have not yet been detected in yeasts and animals (Lusser et al., 1997). In Arabidopsis, HD2 family members have a conserved terminal amino acid region (EFWG motif), and HDT1 and HDT3 contain a C2H2 type zinc finger domain, which may mediate DNA–protein or protein–protein interactions (Zhou et al., 2004).

A total of 18 HDACs have been identified in the genome of Arabidopsis, which belong to three aforementioned HDAC families (Pandey et al., 2002). In recent years, increasing evidences show that HDACs play important roles in regulating epigenetic processes in Arabidopsis in response to abiotic and biotic stresses. HDA6 and HDA19 are involved in abscisic acid (ABA) signaling pathway, and can be induced by jasmonic acid (JA) and ethylene (ET) (Chen et al., 2010; Chen & Wu, 2010). HDA6 can interact with EIN3 and JAZ proteins, and is involved in ET and JA signaling pathways (Zhu et al., 2011), while HDA19 participates in salicylic acid (SA)-mediated plant defense responses and regulates PR1 (Pathogenesis Related 1) expression by inhibiting WRKY38 and WRKY62, which encode two structurally related type III WRKY transcription factors (Kim et al., 2008). In hda19 mutant, genes related to SA signaling pathway have been observed to be overexpressed and PR genes have been demonstrated to show increased expression. Besides, the mutant has also been reported to exhibit enhanced resistance to Pseudomonas syringae pv tomato strain DC3000 (Choi et al., 2012). The HD2D gene has been noted to influence plant growth, development, and abiotic response (Han et al., 2016). In rice, OsHDACs play an important role in abiotic and biotic stresses. Both HDT701 and HDT705 have been observed to be localized in the nucleus and involved in the regulation of seed germination in response to abiotic stresses (Zhao et al., 2014a; Zhao et al., 2016). Overexpression of HDT701 can decrease the level of histone H4 acetylation and enhance susceptibility to the rice pathogens Magnaporthe oryzae and Xanthomonas oryzae, indicating that HDT701 negatively regulates the innate immunity by modulating the levels of histone H4 acetylation of pattern recognition receptor and defense-related genes (Ding et al., 2012).

In maize, ZmHDACs have been reported to regulate plant development, seed dormancy, and germination. Maize HDAC genes HDA101, HDA102, and HDA108 have been found to show similar expression levels in endosperm; for example, HDA108 controls vegetative and reproductive development and HDA101 influences seed development by regulating histone acetylation levels (Varotto et al., 2003; Rossi et al., 2007; Yang et al., 2016; Forestan et al., 2018). In addition, ZmHDACs activities are required for GA-induced programmed cell death in aleurone layers, and regulate programmed cell death via ROS-mediated signal transduction pathway (Hou et al., 2015; Hou et al., 2017). Nevertheless, systematic analysis of all the HDAC genes and their responses to abiotic and biotic stresses in maize has not yet been reported. In the present study, the HDAC gene family was identified from the maize genome, the evolutionary relationships between maize, Arabidopsis, and rice were determined, and the conserve domain and tissue specific and stress responsive expression profiles were further analyzed. Together, our results revealed the importance of ZmHDAC genes in various aspects of plant development and response to abiotic or biotic stress.

Materials and Method

Plant materials and treatments

The maize inbred line B73 was grown in the experimental field in Baoding (Hebei Agricultural University, Hebei province, China). For methyl jasmonate (MeJA; Sigma, USA), SA (Thermo Fisher Scientific, USA) treatments, MeJA (100 µM) and SA (100 µM were sprayed onto the entire seedlings at the V3 growth stage (Li et al., 2015; Wang et al., 2015). Cold and heat stresses were applied by growing the seedlings under control conditions in an incubator at 4 °C or 42 °C from 12:00 pm for 12 h. The control treatment temperature was 25 °C.

Identification of HDACs in maize

The maize genetics and genomics database (MaizeGDB, https://www.maizegdb.org) were searched to identify HDACs using BLASTP with a standard e-value <1e−5, with Arabidopsis and rice HDAC protein sequences as queries. Furthermore, the domains of all protein sequences were analyzed using Hidden Markov Model (HMM) of Pfam (http://pfam.sanger.ac.uk/search) and SMART database (http://smart.embl-heidelberg.de). The HDACs in maize in online server ExPASy were examined using bioinformatics software ExPASy-ProtParam tool (http://web.expasy.org/protparam/).

Phylogenetic analysis

ClustalW with default parameters was employed for multiple alignment of all HDAC protein sequences of maize, rice, and Arabidopsis (Larkin et al., 2007). The results of multiple alignment were imported to MEGA7.0 software for phylogenetic analysis, with neighbor-joining (NJ) method and 1,000 bootstrap re-samplings (Kumar, Stecher & Tamura, 2016).

Conserved domain analysis

Protein sequences of HDAC family genes were obtained from maize genomics database. The conserved domain of maize HDACs was identified by SMART (http://smart.embl-heidelberg.de) and Pfam (http://pfam.xfam.org) with default parameters, and plotted by IBS software (Liu et al., 2015).

Expression analysis of HDACs in maize

Published transcriptome datasets were downloaded from NCBI Short Read Archive database (SRA). The RNA-Seq datasets (Table S2) were mapped to the reference genome of maize (B73 RefGen_V3) using TopHat with default parameters (Trapnell, Pachter & Salzberg, 2009). Cufflinks software was used to calculate the expression levels using default parameters (Trapnell et al., 2012). The gene expression levels were normalized by gene length and read numbers to calculate FPKM (fragments per kilobase of transcript per million mapped reads) values. Heatmaps of maize HDACs expression levels were constructed by using Heml software (Deng et al., 2014).

RNA isolation and quantitative real-time PCR

All samples were homogenized in liquid nitrogen before RNA isolation. Total RNA was isolated using TRIZOL® reagent (Invitrogen, USA) and purified with Qiagen RNeasy columns (Qiagen, Germany). Quantitative real-time PCR (qRT-PCR) was conducted using actin as an internal reference and cDNAs from samples collected at different time points as template with TransStart Tip Green qPCR SuperMix according to the manufacturer’s instructions. Furthermore, comparative Ct analysis (2-ΔΔCt) of each gene in inbred B73 and its relative expression levels at different time points was employed, and quantitative data were expressed as mean ± standard error of mean (SEM). The primer sequences used in this study are listed in Table S1.

Immunoblotting analysis

The nuclei from leaf tissue in maize V3 stage were purified by sucrose density gradient centrifugation, and nuclear proteins were isolated by protein extraction buffer. The nuclear proteins mixed with loading buffer were boiled for 5 min. Then, the proteins were separated by 12% SDS-PAGE and the separated proteins were blotted onto PVDF membranes (Millipore, 0.22 µm). The membranes were blocked (5% milk dissolved in 1 × TBST) at room temperature for 2 h, and incubated overnight at 4 °C with anti-H3 (Abcam, ab1791), anti-H3K9ac (Abcam, ab10812), anti-H4K5ac (Millipore, 07-327), and anti-H4K8ac (Abcam, ab15823) antibodies. The membranes were washed thrice with TBST for 10 min and incubated for 2 h at room temperature with horseradish peroxidase labelled secondary antibodies. The membranes were washed three times with TBST, incubated in ECL for 1 min, and examined using X-OMAT BT film in darkroom.

Results

Identification and phylogenetic analysis of HDACs in maize

The protein sequences of AtHDACs and OsHDACs were used as queries to conduct sequence homology searches against the maize genomics database. The length of complete coding sequence (CDS) and number of amino acids encoding the ZmHDAC family genes were obtained from the maize genomics database. In total, 18 independent maize HDAC family genes were identified (Table 1). The CDS of the retrieved ZmHDAC genes ranged from 501 to 2103 bp, while predicted proteins ranged from 166 to 700 amino acids with calculated molecular weights from 18.91 to 76.54 kDa and isoelectric points from 4.61 to 9.31 (Table 1).

Table 1 HDAC genes information in maize.

	Gene ID	Gene name	Chr	Start	End	CDS (bp)	AA	MW (Da)	pI	
RPD3/ HDA1	GRMZM2G163572		chr5	156416899	156406486	1515	504	56689.26	5.65	
GRMZM2G172883	HDA101	chr4	231342828	231335846	1551	516	57803.5	5.43	
GRMZM2G081474		chr9	84349561	84342406	1410	469	52779.31	5.73	
GRMZM2G119703	HDA102	chr2	59103871	59090885	1092	363	41405.76	4.77	
GRMZM2G136067	HDA108	chr4	65988818	65983218	1377	458	50940.17	5.43	
GRMZM2G367886		chr6	168403680	168402676	501	166	18907.73	9.17	
GRMZM2G456473		chr3	182718225	182648800	1152	383	43341.55	8.91	
GRMZM2G457889	HDA109	chr2	210298363	210311868	2103	700	76544.96	5.54	
GRMZM2G056539		chr8	99989295	99984845	900	299	31909.77	5.33	
GRMZM2G046824		chr8	99109552	99100490	834	277	29263.26	6.1	
GRMZM2G107309	HDA110	chr7	7285361	7273053	1860	619	66186.64	5.74	
GRMZM2G008425		chr9	80424376	80418132	1056	351	38784.55	6.71	
SIR2	GRMZM2G058573	SRT101	chr10	125947058	125922937	1314	437	49050.03	9.31	
GRMZM5G807054		chr10	10823442	10811717	1056	351	39188.5	8.91	
HD2	GRMZM2G057044	HDT101/HDA106	chr3	164420298	164429357	882	293	31810.94	4.29	
GRMZM2G100146	HDT102/HDA103	chr8	135667488	135662882	906	301	32402.15	4.61	
GRMZM2G159032	HDT103/HDA105	chr6	161429635	161426908	903	300	32501.43	4.69	
GRMZM5G898314	HDT104	chr8	162666204	162662841	858	285	30435.21	4.75	
Notes.

Start The first physical position of the gene on the chromosome

End The last physical position of the gene on the chromosome

CDS Coding sequence

AA Amino acid length

MW molecular weight

pI Isoelectric point

To evaluate the evolutionary relationship of plant HDACs, phylogenetic analysis was performed using the protein sequences of HDACs from maize, rice, and Arabidopsis. The phylogenetic tree indicated that the 18-uncovered maize HDACs could be grouped into three types characterized by distinctive protein structures (Fig. 1). In maize, RPD3/HDA1 family HDACs consisted of 12 members based on their sequence similarity, all of which exhibited a characteristic HDAC domain (Fig. 2) and could be further divided into three classes based on sequence similarity. Class I, II, and III included 6 HDACs, 5 HDACs, and 1 HDAC, respectively (Fig. 1). The phylogenetic analysis also demonstrated that maize has two SIR2 family HDACs with highly conserved Sir2 domains. Finally, four plant-specific HDACs (HD2 family) were identified, which might indicate that this protein has high DNA-binding affinity or could mediate protein–protein interactions.

Figure 1 Phylogenetic relationship of HDAC gene family among maize, rice, and Arabidopsis.

Multiple sequences alignment and phylogenetic tree construction were performed using MEGA7.0. The value at the nodes represents bootstrap values from 1,000 replicates. Different groups are shown by different colors.

Figure 2 Phylogenetic analysis and domain architecture of ZmHDACs.

Construction of phylogenetic tree based on ZmHDACs amino acid sequences. Conserved domains of ZmHDACs were identified by Pfam and SMART. Different domains are indicated by different colors. The lengths of the domains in each protein are proportional. The NJ phylogenetic tree of ZmHDACs protein sequences was constructed using 1,000 bootstrap replicates by MEGA 7.0. HDAC, Histone deacetylase domain; Sir2, Sir2 catalytic domain; ZnF_C2H2, Zinc finger C2H2 type domain.

Tissue-specific expression profiles of HDAC genes in maize

To investigate tissue- or organ-specific expression profiles of HDAC genes in maize, transcriptome data from the SRA database were used, which included 22 different tissues or organs (Sekhon et al., 2014). As indicated in Fig. 3, the expression patterns of the maize HDAC genes could be divided into three clusters, cluster 1, cluster 2, and cluster 3. Cluster 1 comprised two subgroups, C1-Sub1 and C1-Sub2, according to their expression levels. Genes in C1-Sub1 had a low expression level in anther and pollen. In contrast, genes in C1-Sub2 were highly expressed in anther and pollen, presenting an opposite expression trend, which indicated that these genes might be associated with reproductive growth. Genes in cluster 2 exhibited a low expression level in all the tissues investigated, especially GRMZM2G367886, GRMZM2G456473, and GRMZM5G807054. Genes in cluster 3 showed a higher expression level in root, stem, shoot apical meristem (SAM), seed, and endosperm, and had a lower expression level in anther and pollen, suggesting the possible involvement of these genes in cell differentiation and seed development.

Figure 3 Heatmaps representing the expression profiles of ZmHDAC genes in several tissues.

Tissue-specific expression patterns of ZmHDAC genes associated with 22 different tissues or organs. The color scale on the right indicates expression values, with blue denoting high expression level and yellow representing low expression level.

Expression profiles of HDAC genes under biotic stress

To explore the potential roles of ZmHDACs in biotic stress responses, we analyzed the expression pattern of HDAC family genes under Fusarium verticillioides infection from the public expression profile datasets (Shu et al., 2017). The results showed that the expression level of HDAC family genes presented different trends under F. verticillioides infection (Fig. 4). For example, the expression levels of GRMZM2G057044 gradually decreased with the increase in treatment time. The expression level of GRMZM2G163572 was down regulated at 4 and 72 hpi (hours post infection), and up regulated at 48 hpi. The expression levels of GRMZM2G457889 were relatively stable before 48 hpi, but decreased at 72 hpi. In addition, GRMZM2G456473 was up regulated at 12 hpi, while GRMZM2G081474 was down regulated at 24 hpi.

Figure 4 Expression patterns of ZmHDAC genes under F. verticillioides infection.

The color scale on the right indicates fold changes (inoculated/mock), with blue denoting high fold change and yellow showing low fold change.

Expression profiles of HDAC genes under abiotic stress

It has been indicated that HDACs play important functions in response to abiotic stresses (Makarevitch et al., 2015). To explore the potential roles of ZmHDACs in abiotic stress responses, we treated maize seedlings with MeJA and SA, which are the most important stress-protective phytohormones, to determine the expression levels of ZmHDACs under hormones treatment. As shown in Fig. 5, ZmHDAC genes significantly responded to MeJA treatments. Three HD2 family genes (GRMZM5G898314, GRMZM2G100146, and GRMZM2G159032) and three Class I family genes (GRMZM2G163572, GRMZM2G136067, and GRMZM2G367886) were downregulated. However, the expression levels of GRMZM2G172883, GRMZM2G046824, GRMZM2G056539, and GRMZM2G456473 were induced in 3 and 6 h, but gradually decreased after 6 h of MeJA treatment. Under SA treatment (Fig. 6), most of the ZmHDAC genes were downregulated, except a Class II family gene GRMZM2G046824, which was induced in 3 and 6 h of SA treatment. With regard to the downregulated genes under SA treatment, a fraction of the genes, such as GRMZM2G172883, GRMZM2G119703, GRMZM5G807054, GRMZM2G898314, GRMZM5G100146, and GRMZM2G057044, were downregulated in 3 and 6 h and upregulated in 12 and 24 h of SA treatment. These results suggested that differentially expressed genes might be involved in JA and SA signaling pathways.

Figure 5 Expression patterns of ZmHDAC genes in seedling leaf under MeJA treatment.

(A–R) Expression pattern of each HDAC gene under SA treatment from 0 h to 24 h. qRT-PCR was performed using gene-specific primers. Data are the mean ± SEM of three independent experiments.

Figure 6 Expression patterns of ZmHDAC genes in seedling leaf under SA treatment.

(A–R) Expression pattern of each HDAC gene under MeJA treatment from 0 h to 24 h. qRT-PCR was performed using gene-specific primers. Data are the mean ± SEM of three independent experiments.

Furthermore, we analyzed the transcriptome data under heat, UV, cold, salt, and drought stresses. The findings indicated that the maize HDAC genes presented significant differential expression under abiotic stress (Fig. 7). Under drought stress, most of the genes, such as GRMZM5G898314 and GRMZM2G119703, were significantly upregulated, while GRMZM2G457889 was downregulated. Similarly, under heat stress, most of the genes were downregulated, while GRMZM2G807054 and GRMZM2G172883 were upregulated. In addition, we analyzed the status of H3 acetylation following cold and heat treatments. The levels of H3K9ac and H4K5ac decreased under cold treatment (Fig. 8), but significantly increased after heat treatment, when compared with those in plants grown at normal temperatures. These findings suggested that maize HDACs might be involved in abiotic stress responses by regulating histone acetylation levels.

Figure 7 Expression patterns of ZmHDAC genes under heat, cold, salt, UV, and drought stresses.

The color scale on the right indicates expression values, with blue denoting high expression level and yellow representing low expression level.

Figure 8 Immunoblot for the detection of H3K9ac and H4K5ac levels in seedling of the B73 inbred line under cold and heat treatments.

Immunoblot with anti-H3 antibody was used as a loading control. CK mean the control treatment temperature (25°C), cold and heat represented 4°C 381 and 42°C treatment.

Discussion

HDACs, also known as lysine deacetylases (HDACs), are a class of enzymes that remove acetyl groups from core histones (H2A, H2B, H3, and H4), thereby regulating gene expression (Makarevitch et al., 2015), and play critical roles in genome stability, plant growth and development, and response to environmental stresses (Luo et al., 2012a; Liu et al., 2014). Genome-wide identification and characterization of HDACs have been reported in several plant species, such as Arabidopsis, Solanum lycopersicum, and Oryza sativa (Pandey et al., 2002; Fu, Wu & Duan, 2007; Zhao et al., 2014b). In maize, although HDAC genes have been identified (Demetriou et al., 2009), systematic analysis of the HDAC gene family in response to abiotic and biotic stresses is limited. In this study, 18 HDAC genes were identified in the maize genome, which belonged to three subfamilies: Class I (6), Class II (5), Class III (1), SRT (2), and HD2 (3). The number of ZmHDAC gene family in maize was found to be almost consistent with that in Arabidopsis, rice, and tomato, and gene expansion of HDACs was not obvious in maize (Pandey et al., 2002; Fu, Wu & Duan, 2007; Zhao et al., 2014b). In Class I, Class II, and Class III family, all members had a conventional HDAC conserved domain, and Sir2 and ZnF_C2H2 domains were identified in SRT and HD2 families. Two novel typical HDAC family members and one SRT family member were identified in the present study, which are important for research on HDACs in maize (Demetriou et al., 2009). The results of conserve domain analysis indicated that the function of HDAC family members in different plant species might have been stable during evolution, such as GRMZM2G136067 (HDA108) that can functionally complement a yeast rpd3 null mutant and influence the characteristic acetylation pattern on H4, suggesting a possible function for GRMZM2G136067 in the histone deposition process (Rossi, Hartings & Motto, 1998; Kolle et al., 1999).

In plants, HDACs are the key regulators of histone modification and chromatin remodeling, implying that epigenetic regulation play an important role in controlling gene expression in the developmental stages and responses to abiotic stress (Luo et al., 2012a; Ma et al., 2013; Liu et al., 2014). Accordingly, transcriptome data from public databases were further explored in the present study to dissect the expression profiles of the ZmHDAC genes. Several previous studies have shown that HDA6 is involved in plant growth and development, such as leaf development and flowering (Probst et al., 2004; Yu et al., 2011). GRMZM2G136067, an ortholog of HDA6, exhibited an expression pattern similar to GRMZM2G119703, with higher transcript accumulation during endosperm development (Varotto et al., 2003), suggesting that GRMZM2G136067 and GRMZM2G119703 might be involved in endosperm development. GRMZM2G172883, an ortholog of HDA19, was highly expressed during seed development and germination, and several studies have confirmed that this gene plays an important role in the regulation of histone deacetylation during seed development (Rossi et al., 2007; Yang et al., 2016). The expression of GRMZM2G046824 and GRMZM2G56539, orthologs of HDA8 involved in sperm cell formation, was higher in anther and pollen, revealing that these two genes might be involved in flower development in maize. However, the tissues covered by the current datasets were limited, and tissue-specific analysis of ZmHDACs must include more types of tissues or datasets.

HDA6 and HDA19 can be induced by JA and ET (Zhou et al., 2005; Chen & Wu, 2010). HDA6 can interact with EIN3 and JAZ proteins, and is involved in ET and JA signaling pathways (Zhu et al., 2011). HDA19 participates in SA-mediated plant defense responses and regulates PR1 expression by inhibiting the activity of WRKY38 and WRKY62 (Kim et al., 2008). The results of the present study revealed that several genes were induced in 3 and 6 h, but gradually decreased after 6 h of MeJA treatment. With regard to SA treatment, several genes were gradually downregulated at 3 and 6 h, and then upregulated at 12 and 24 h after the treatment. Disease resistance is regulated by several signal transduction pathways in which SA and JA function as key signaling molecules. In our results, several genes were induced or suppressed by JA and SA, which indicated that these genes might be involved in plant–pathogen interaction. For example, the expression pattern of GRMZM2G172883, an ortholog of HDA19 that can be induced by JA (Zhou et al., 2005), was similar to that noted in Arabidopsis. In addition, the roles of HDA19 in plant defense responses insinuated that GRMZM2G172883 might have some functions to regulate plant immunity. These findings implied that the above-mentioned genes might be involved in hormone signaling pathways and plant defense responses.

Several studies have demonstrated that HDACs are involved in plant resistance to abiotic stress (Yuan et al., 2013). For instance, it has been reported that Arabidopsis overexpressing HD2C exhibited greater tolerance to salt and drought stresses than the wild-type plants (Sridha & Wu, 2006). Besides, HD2D can confer tolerance to abiotic stresses, including drought, salt, and cold stresses in Arabidopsis (Han et al., 2016), whereas HD2C can interact with HDA6 and regulate ABA-responsive gene expression by histone deacetylation (Luo et al., 2012b). In rice, several RPD3-type HDACs have been reported to be repressed under high salt and drought treatments (Hu et al., 2009). In the present study, under salt stress, most of the HDAC genes were downregulated, while GRMZM2G056539 and GRMZM2G172883 were upregulated. A similar phenomenon also occurred under heat treatment, with most of the HDAC genes being downregulated. It has been demonstrated that the expression of HDACs and the levels of histones H4ac, H3K9ac, and H4K5ac were reduced by low temperature in maize (Hu et al., 2011). As a considerable number of HDAC genes were downregulated under heat stress, we used immunoblotting analysis to demonstrate the changes in histone acetylation levels under heat stress. The expressions of histones H3K9ac and H4K5ac were reduced under cold stress, whereas the histone acetylation levels of these genes were increased under heat treatment. These results indicated that histone acetylation may play pivotal roles in plant responses to both cold and heat stresses.

Conclusion

In summary, we performed comprehensive analyses of HDAC gene family in maize, and identified 18 HDAC genes that can be divided into RPD3/HDA1, SIR2, and HD2 families. The domain arrangement was considerably conserved among members in the same groups or subgroups. Some ZmHDAC genes showed significant tissue-specific expression, as noted in tissue-specific expression profiles, suggesting that these genes might participate in different organ development. Several HDAC genes exhibited different expression trends following MeJA treatment, SA treatment, and fungal infection, indicating that these genes might be involved in defense response. Interestingly, our results indicated that global histone acetylation in maize was affected by heat stress, revealing that HDACs might be involved in the response of maize to abiotic stress. These findings suggest that ZmHDACs might be important for plant development and response to biotic and abiotic stress. Nevertheless, further research is required to determine the function and molecular mechanisms of ZmHDACs in plant responses to biotic and abiotic stresses, which could provide tools for the improvement of maize productivity.

Supplemental Information

Table S1 Primers used in this study

Click here for additional data file.

Table S2 Public RNA-Seq datasets of different tissues used in Fig. 3

Click here for additional data file.

Supplemental Information 1 Raw data for gene expression

Click here for additional data file.

Supplemental Information 2 Raw data for Fig. 8

Click here for additional data file.

Additional Information and Declarations

Competing Interests

Author Contributions

Data Availability

The authors declare there are no competing interests.

Kang Zhang conceived and designed the experiments, performed the experiments, analyzed the data, prepared figures and/or tables, authored or reviewed drafts of the paper, and approved the final draft.

Lu Yu performed the experiments, analyzed the data, prepared figures and/or tables, and approved the final draft.

Xi Pang and Hongzhe Cao performed the experiments, prepared figures and/or tables, and approved the final draft.

Helong Si and Jinping Zang analyzed the data, prepared figures and/or tables, and approved the final draft.

Jihong Xing and Jingao Dong conceived and designed the experiments, authored or reviewed drafts of the paper, and approved the final draft.

The following information was supplied regarding data availability:

The raw measurements are available in the Supplementary Files.

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
