# Peer review of "In silico analysis of maize HDACs with an emphasis on their response to biotic and abiotic stresses"

_PeerJ, doi:10.7717/peerj.8539_

## Round 0.1 · original submission · Major Revisions

Dear Dr Zhang
Both reviewers and myself were interested by the topic of this manuscript. However, I send you here a number of comments raised by two reviewers. Please consider if you would be able to improve the manuscript in function of all the comments raised by them. Specially comments regarding validity of the findings, experimental design and readability and presentation of the manuscript. If you are able to modify the manuscipt while improving its final readability I will be able to send it again for review.
Best regards

Reviewer 1 ·

Basic reporting

1. English must be significantly improved throughout the manuscript. There are multiple grammar and spelling errors as well as unclear or complicated sentences that difficult the comprehension of the manuscript as well as the review process. Some examples are in lines 40, 43, 51, 74-77, 126-127, 234-235, 240-241...(but there are more).
2. The introduction needs more details, specially when describing the existing knowledge about HDACs in maize (lines 67-71). There are also missing works that could enrich this paragraph, including the one by Forestan C et al 2018 as an example.
3. Authors should also consider to better explain/detailed the existing knowledge to reach a more widespread audience. For example, what is PR1, WRKY38, WKRY62 (line 57) or Pst DC3000 (line 59)? This comment should be considered along the manuscript.
4. Acronyms should be defined the first time they are used. JA, SA, MeJA, ABA, etc.
5. Lines 73-75. What do the authors mean with “…characterization of all HDAC genes…”? This sentence could be interpreted as if there is no knowledge about HDACs in maize at all… please revise.
6. Please revise figure legend of Figure 2.
7. Line 186: “…hpi after infection…” sounds redundant.
8. Please revise sentences in lines 213-215.
9. Line 234: what do the authors mean with “…was almost consistent…”?
10. Line 235: what do the authors mean with “…gene expression of HDACs were not obvious in maize…”?
11. Lines 240-241: what do the authors mean with “…can deacetylate the characteristic deacetylation pattern on H4…”?

Experimental design

12. An HDAC phylogenetic analysis including several HDAC proteins from A. thaliana, O. sativa and Z. mays has been already performed (Demetriou et al 2009). This work should be discussed. In what the presented manuscript differs from previous work? What do the authors consider that this new analysis brings to light?
13. It is unclear how authors proceeded for the identification of maize HDACs. Did authors search for each described At and Os HDAC homologs by sequence BLAST? If yes, please provide a table showing the hits obtained in each BLAST search, including the e-values, score, etc, instead of or in addition to Table 1.

Validity of the findings

14. Accession numbers of the transcriptomic data used for the different analyses should be provided, as well as their corresponding reference.
15. In literature, it is mentioned that there is 15 HDACs in maize (examples are Hu Y et al 2011; Demetriou K et al. 2009). Please comment on this. Which are the two new HDACs that authors have found? Also, it would be useful to indicate the commonly used (and already published) HDAC names on top of the systematic names used along the manuscript (common names of putative orthologs are only mentioned in the discussion for some genes). This would significantly facilitate to relate this work with previous publications.
16. Lines 216-219: Authors claim that cold treatment causes a decrease in H3K9ac. However, Figure 8 shows no effect. Also, the other effects on histone acetylation are quite mild. How many times was this experiment done? Are these results reproducible?

Additional comments

The manuscript presented by Kang Zhang et al. describes the effect of biotic and abiotic stresses on the expression of histone deacetylases genes in maize. Using existing transcriptomic data and their own generated expression analyses by RT-qPCR, authors describe the changes in the expression of HDAC genes upon different stresses. The generated analyses contribute to our understanding of how HDAC genes are transcriptionally regulated in maize. However, several improvements of the presented manuscript are needed.

Reviewer 2 ·

Basic reporting

The results are interesting but limited. The study is preliminary and descriptive. No underlying mechanism was provided. The conclusions are ambiguous. Based on the present limited findings it shall be difficult to claim such conclusions. The conclusions are not supported by substantial evidences. And the figures are not easily readable. Figure legends are confusing and unclear. Modify the figure legends.

Experimental design

Materials & Methods need to be improved.

Validity of the findings

The study is preliminary and descriptive. The conclusions are not supported by substantial evidences.

Additional comments

Manuscript ID: peerJ-reviewing-38864-v0

The present investigation revealed the significant role of ZmHDACs in plant development and response to biotic and abiotic response to stresses. Analyses of HDAC gene family by using In silico protein and gene expression analysis. HDAC gene family can be divided into 17 genes, which can be further categorized into three subfamilies, including RPD3/HDA1, SIR2, and HD2. Tissue-specific genes were revealed. GRMZM2G046824, GRMZM2G56539, and GRMZM2G008425 were highly expressed in anther and pollen, while GRMZM2G100146 and GRMZM2G159032 were highly expressed in seed and endosperm. Furthermore, MeJA, SA, fungal, and heat treatment has revealed the role of HDACs genes in maize.
The results are interesting but limited. The study is preliminary and descriptive. No underlying mechanism was provided. The study is preliminary and descriptive. The conclusions are not supported by substantial evidences. The present conclusions include following questions:
• On what basis HDACs were divided into 12 members? How they are categorized? Is it members or categories?
• How do authors interpret that the genes present in higher levels in the specific tissues are involved in their functionality?
• How have authors categorized higher expression and lower expression of genes? What value was designated as higher and lower expression levels?
• Why have results not been verified and discussed in details?
• How do authors interpret and discuss the Genes in Cluster3 which have shown a lower expression level in all tissues investigated?

English Level: The English used in the article is not readable and not good enough to convey the scientific meaning correctly. There are several spelling and grammatical mistakes incorporated throughout the manuscript. To enhance the quality and clarity of manuscript, it requires a thorough revision from someone with full professional proficiency. Gene terminologies should be italicized.
For example:
Introduction Line 43: Members of the SIR2 family are conservative from prokaryotic to eukaryotic
Modified to: “conservative” should be modified to “conserved”.
Line 46-49: Needs to be modified
Line 231: Needs to be modified
Overall Recommendation: Reconsider after major English language revision.
Title
The title of the manuscript is broad and misleading. The title should be restricted specifically to the limited findings.
Abstract
Add the principal results and major conclusions clearly. Revise it and add specific data and
results of the present work.
Introduction
Good introduction is succinct. The importance and usefulness of this research work is not
clear. Add it.
Add more related and relevant references. Add additional references, such as:
Hou H, Zheng X, Zhang H, Yue M, Hu Y, Zhou H, Wang Q, Xie C, Wang P, Li L. Histone Deacetylase Is Required for GA-Induced Programmed Cell Death in Maize Aleurone Layers. Plant Physiol. 2017;175(3):1484-1496. doi: 10.1104/pp.17.00953.
Hou H, Wang P, Zhang H, Wen H, Gao F, Ma N, Wang Q, Li L Histone acetylation is involved in gibberellin-regulated sodCp gene expression in maize aleurone layers. Plant Cell Physiol; 2015; 56: 2139–2149
Forestan C., Farinati S., Rouster J., Lassagne H, Lauria M., Ferro N., D.,,and Serena Varotto Control of Maize Vegetative and Reproductive Development, Fertility, and rRNAs Silencing by HISTONE DEACETYLASE Genetics; 2018; 108. Apr; 208(4): 1443–1466. doi: 10.1534/genetics.117.300625

Materials & Methods
Which protocol was used for JA and SA treatments in plants? Or is it a new protocol designed by the authors? Explain it.
Why only one concentration of JA (100 μM ) and SA (100 μM ) were used for the treatment of the maize plant? Why no other concentrations were tested?
Which statistical analysis was used to demonstrate the specific data? Add in the Methodology.
Discussion
In-depth discussion is missing. The Discussion should explain the significance of the results and place them into a broader context. It is often helpful to the reader to indicate the directions in which the work might be built on going forward. It should not be redundant with the Results. But in the present manuscript, this section has mainly presented by the results. There are more relevant references presents.
Modify the discussion section thoroughly.
Discussion section seems to be the repetition of results. For example:
Line 260-266: “In our results, GRMZM2G172883, GRMZM2G046824, GRMZM2G056539,
and GRMZM2G456473 were induced in 3 h and 6 h, and the expression levels of these genes
were gradually decreased after 6 h MeJA treatment. In SA treatment, GRMZM2G172883,
GRMZM5G807054, GRMZM2G898314 and GRMZM2G100146 were gradually down-regulated
at 3 h and 6 h, and then up-regulated at 12 h and 24 h. GRMZM2G172883, ortholog of AtHDA19, showed the similar expression pattern in Arabidopsis. These results suggested that these genes might be involved in hormone signaling pathways and plant defense responses.”
It is not clear. Modify it: “The number of ZmHDAC gene family in maize was almost consistent with those in Arabidopsis, rice and tomato, and gene expression of HDACs were not obvious in maize (Fu et al. 2007; Pandey et al. 2002; Zhao et al. 2014b).”
It is not clear. Modify it: Two members of class II family GRMZM2G046824, GRMZM2G56539
and one members of class II family GRMZM2G008425 were higher expressed in anther and
pollen, suggesting that these genes might be involved in flower development.
Conclusion
Why the future perspective was not mentioned? Describe it and add it in the conclusion section.
Figure Legends
The figures are not easily readable. Figure legends are confusing and unclear. Modify the figure legends.

---

## Round 0.2 · Major Revisions

Dear authors

Referees raised several concerns. The novelty of the findings is not going to be taken into account (it is not the journals policy) but the readability and required corrections to the English will. Please review the manuscript addressing point by point the comments raised and I will take the decision later, but I want to make sure all referees comments are addressed, or discussed in a convincing way, in the letter to referees, and specially that the English and style in general is improved so that the information can be easily comprehended by readers.

Reviewer 1 ·

Basic reporting

This is a revised version of the manuscript presented by Kang Zhang et al. Most of my comments have been appropriately addressed. Nevertheless, there are still some points that might require attention.

1. Despite English improvement throughout the manuscript, there is still some typos and grammar errors to be corrected (examples include the use of caps in “Acetylation” (line 50), space between “plant” and “growth” in line 246), “diacetylation” (line 264) and grammar of the sentence in lines 184-186).
2. Reference to Figure 2 in line 183 seems wrong. The different classes of HDAC are only depicted in Figure 1.

Experimental design

No comment

Validity of the findings

3. Even if highly reproducible, the effect of cold stress on acetylation of H3K9 is not convincing, especially because western blot is a quantitative limited technique and the claimed effect is extremely mild. Authors should consider concluding about this result with more caution.

Additional comments

Suggestion: If possible, it would be useful for the readers to represent Figure 3 in such a way that the clusters and subgroups mentioned in lines 191-201 are labeled within the figure.

Reviewer 2 ·

Basic reporting

Major English revisions are required.

Experimental design

Methodology: What temperature was used for cold shock or for heat treatment stress and for what time period? Authors need to mention it at appropriate sections in the manuscript.

Validity of the findings

Novelty of the present investigation raises question. As previously it has been determined 18 HDACs in Arabidopsis and Rice. These HDACs are divided into three families. In the present study the authors have used in silico methodologies to determine similar 18 HDACs in maize (abstract Line 31, L32) and in the present study also HDACs were divided into three families. In fact, similar results to response of HDACs to JA and SA and abiotic stresses were obtained. The present data is limited.

Additional comments

Manuscript ID: #38864
New Comments are in “red”.
Previous comments are in “black”.
Authors response to previous comments is in “blue”.
Novelty of the present investigation raises question. As previously it has been determined 18 HDACs in Arabidopsis and Rice. These HDACs are divided into three families. In the present study the authors have used in silico methodologies to determine similar 18 HDACs in maize (abstract Line 31, L32) and in the present study also HDACs were divided into three families. In fact, similar results to response of HDACs to JA and SA and abiotic stresses were obtained. The present study is limited.
Methodology: What temperature was used for cold shock or for heat treatment stress and for what time period? Authors need to mention it at appropriate sections in the manuscript.
Italicized the gene terminology. For example, HDAC (L96, L98, L270, L282, L321, L366, L371, L375, L379, L383), ZmHDAC (L94, L225, L254), etc.
Recommendation: Major English revisions need to be done by a Native English Speaker. Mention what was actually modified in point-by-point response to previous and new comments.
For the following responses to previously raised queries, authors need to mention what was actually modified in the revised version of the manuscript and also mention the line numbers and page numbers to indicate those modifications.
Q1. Why have results not been verified and discussed in details?
Answer: Thank you for your question. All results have been verified and discussed in the revised manuscript.
Q2. The title of the manuscript is broad and misleading. The title should be restricted specifically to the limited findings.
Answer: Thank you for your suggestion. The title has been modified.
Q3. Add the principal results and major conclusions clearly. Revise it and add specific data and results of the present work.
Answer: Thank you for your comments. The abstract has been improved in the revised manuscript.
Q4. Which statistical analysis was used to demonstrate the specific data? Add in the Methodology.
Answer: Thank you for your helpful suggestion. Statistical analysis has been added to the method.
Q5. In-depth discussion is missing. The Discussion should explain the significance of the results and place them into a broader context. It is often helpful to the reader to indicate the directions in which the work might be built on going forward. It should not be redundant with the Results. But in the present manuscript, this section has mainly presented by the results. There are more relevant references presents.
Answer: Thank you for your helpful advice. The discussion has been improved in the revised manuscript.
Q6. Modify the discussion section thoroughly.
Discussion section seems to be the repetition of results. For example:
Line 260-266: “In our results, GRMZM2G172883, GRMZM2G046824, GRMZM2G056539, and GRMZM2G456473 were induced in 3 h and 6 h, and the expression levels of these genes were gradually decreased after 6 h MeJA treatment. In SA treatment, GRMZM2G172883, GRMZM5G807054, GRMZM2G898314 and GRMZM2G100146 were gradually down-regulated at 3 h and 6 h, and then up-regulated at 12 h and 24 h. GRMZM2G172883, ortholog of AtHDA19, showed the similar expression pattern in Arabidopsis. These results suggested that these genes might be involved in hormone signaling pathways and plant defense responses.”
Answer: Thank you for your advice. The sentence has been modified.
Q7. It is not clear. Modify it: “The number of ZmHDAC gene family in maize was almost consistent with those in Arabidopsis, rice and tomato, and gene expression of HDACs were not obvious in maize (Fu et al. 2007; Pandey et al. 2002; Zhao et al. 2014b).”
Answer: Thank you for your advice. The sentence has been modified.
What do authors mean by “was not obvious in maize”? Explain it.
“The number of ZmHDAC gene family in maize
was found to be almost consistent with that in Arabidopsis, rice, and tomato, and gene expansion of HDACs was not obvious in maize (Pandey et al. 2002; Fu et al. 2007; Zhao et al. 2014b).”
Q8. It is not clear. Modify “Two members of class II family GRMZM2G046824, GRMZM2G56539 and one members of class II family GRMZM2G008425 were higher expressed in anther and pollen, suggesting that these genes might be involved in flower development”.
Answer: Thank you for your advice. The sentence has been modified.
Q9. Why the future perspective was not mentioned? Describe it and add it in the conclusion section.
Answer: Thank you for your advice. The future perspective has been added in the conclusion section.
Q10. The figures are not easily readable. Figure legends are confusing and unclear. Modify the figure legends.
Answer: Thank you for your advice. The figure legends have been modified in the revised manuscript.
Not satisfied with the following responses:
Q11. How do authors interpret and discuss the Genes in Cluster3 which have shown a lower expression level in all tissues investigated?
Answer: Thank you for your question. Genes in Cluster 3 showed lower expression level in all tissues. We presume that the tissues covered by the current RNA-Seq datasets are limited. These genes might be expressed in some specific tissues.
Why the authors are not able to explain the lower expression in Cluster 3?
Q12. Why only one concentration of JA (100 μM ) and SA (100 μM ) were used for the treatment of the maize plant? Why no other concentrations were tested?
Answer: Thank you for your comments. The concentrations of JA (100 μM) and SA (100 μM) had also been applied to many other plants, such as Arabidopsis, rice, tomato, wheat, poaceae, soybean, etc. (Chen et al. 2014; Li et al. 2015; Shalmani et al. 2019; Van Hove et al. 2014; Wang et al. 2017; Wang et al. 2015). In literature, HDA19 was induced by JA treatment, and in the present study, HDA101 (GRMZM2G008425, ortholog of HDA19) was induced by JA treatment and repressed by SA treatment. Hence, we presumed that 100 μM JA and100 μM SA were reliable.
New comments to specific sections of the revised manuscript are as follows:
Title
Title needs to be modified. Suggested title: “In silico analysis of maize HDACs with an emphasis on their response to biotic and abiotic stresses”
Introduction
L50: Modify “Acetylation” into “acetylation”
L51: Modify “it was one of the first discovered to influence”
L87: What is PRR?
L90: If it is maize HDAC genes then it is obvious that they are found in the specific maize tissues. No need to repeat maize again and again. For example, “maize endosperm” (L91), maize vegetative (L91), maize aleurone layer (L95), etc. Modify it.
L92: Modify “HDA101 influencing”
Result
L195-L196: Modify “which indicated the associated of these genes”
L211-212: How do authors obtained “The expression levels of GRMZM2G457889 and GRMZM2G107309 were higher before 48 hpi, but suddenly decreased at72 hpi.” The result is ambiguous.
L222, L236: What do authors mean by using the term “obviously downregulated”? Explain and modify it.
Discussion
L245-246: Modify the statement and also correct the spelling mistake “HDACs play critical roles in histone acetylation, which is a improtant part of histone modification (Lee & Workman 2007).”
L246-L248: Incomplete sentence “HDACs have multiple functions in plant growth and development, response to environmental stresses by regulating gene activities through histone
deacetylation (Luo et al. 2012a; Liu et al. 2014.” Modify it.
L255-256: How do authors claim that “gene expression of HDACs was not obvious in maize”? Explain it.
L258-L259: Modify “in relation to previous study”.
L280: What do authors mean by the term “sperm cell germination”? Explain it.
Conclusion
L321: What do authors mean by “Partial ZmHDAC genes? Modify it.
L327-328: Modify “vital to plant development, regulating biotic and abiotic stress response”.

---

## Round 0.3 · Minor Revisions

Dear authors,

Julin Maloof (the Section Editor) has commented and feels that it still needs another round of revision. Specifically, he had the following comments that I would need you to address:

“ Figure 3: The dendogram denoting the clusters is too compressed, it needs to be replotted in a way that the branching pattern can be seen.

Figure 4: Same comment as Figure 3 regarding the dendogram. In addition, this analysis needs a control (0 hour time point and mock treated across the time course)

In Figure 8 what does the abbreviation "CK" stand for? Needs to be defined in the figure legend.

Why is there a supplemental file " Marine_Gastrotrichs_of_Brazil.kml"?

line 126 "conservative" should be "conserved".

The dataset for the Fursarium RNAseq is not listed in Table S2

The datasets for the stress RNAseq (Figure 7) are not listed in Table S2

At least some of the RNAseq data sets have been published but those studies are not cited. Example https://www.ncbi.nlm.nih.gov/pmc/articles/PMC3634062/

---

## Round 0.4 · accepted · Accept

Thanks for performing the required changes and congrats on your publication!